# The microRNA Lifecycle in Health and Cancer

**DOI:** 10.3390/cancers14235748

**Published:** 2022-11-23

**Authors:** Laura Adriana de Rooij, Dirk Jan Mastebroek, Nicky ten Voorde, Elsken van der Wall, Paul Joannes van Diest, Cathy Beatrice Moelans

**Affiliations:** 1Department of Pathology, University Medical Center Utrecht, Utrecht University, Heidelberglaan 100, 3584 CX Utrecht, The Netherlands; 2Department of Medical Oncology, University Medical Center Utrecht, Utrecht University, Heidelberglaan 100, 3584 CX Utrecht, The Netherlands

**Keywords:** microRNA, biogenesis, post-translational regulation, cancer, genetic alterations, biomarkers

## Abstract

**Simple Summary:**

MicroRNAs (miRNAs) regulate messenger RNA (mRNA) expression at the post-transcriptional level. They play an important role within physiological and pathological cellular processes by regulating an estimated 60% of all protein-coding genes. miRNAs are produced quickly through canonical or non-canonical pathways involving many steps and proteins. In cancer, these steps can be altered to promote tumour formation and progression. These aberrations can occur at the gene level, or can be induced during processing and regulation of the target mRNA. This review provides an overview of the current knowledge on the lifecycle of miRNAs in health and cancer. Understanding miRNA function and regulation is essential prior to potential future application of miRNAs as cancer biomarkers.

**Abstract:**

MicroRNAs (miRNAs) are small non-coding RNAs of ~22 nucleotides that regulate gene expression at the post-transcriptional level. They can bind to around 60% of all protein-coding genes with an average of 200 targets per miRNA, indicating their important function within physiological and pathological cellular processes. miRNAs can be quickly produced in high amounts through canonical and non-canonical pathways that involve a multitude of steps and proteins. In cancer, miRNA biogenesis, availability and regulation of target expression can be altered to promote tumour progression. This can be due to genetic causes, such as single nucleotide polymorphisms, epigenetic changes, differences in host gene expression, or chromosomal remodelling. Alternatively, post-transcriptional changes in miRNA stability, and defective or absent components and mediators of the miRNA-induced silencing complex can lead to altered miRNA function. This review provides an overview of the current knowledge on the lifecycle of miRNAs in health and cancer. Understanding miRNA function and regulation is fundamental prior to potential future application of miRNAs as cancer biomarkers.

## 1. Introduction

MicroRNAs (miRNAs) are small non-coding RNAs of ~22 nucleotides that regulate gene expression at the post-transcriptional level [1,2]. They are integrated into the miRNA-induced silencing complex (miRISC) after association with an Argonaute (AGO) protein [3]. In this context, they typically bind to the 3′ UTR of messenger RNAs (mRNAs) with their seed sequence, spanning 2–8 nucleotides from the 5′ end [4]. This binding usually leads to either translational inhibition or degradation of the mRNA, depending on the degree of complementarity between the two RNAs [1]. miRNAs are estimated to regulate around 60% of all protein-coding genes, with an average of 200 targets per miRNA [5,6]. This indicates their important regulatory function in different physiological and pathological cellular processes. The biogenesis and function of miRNAs is therefore a very tightly regulated process, and dysregulation of miRNA production, availability and target regulation has been associated with various human diseases, including cancer [7].

The role of miRNAs in cancer has been extensively researched. They have been found to be dysregulated in many cancer types and stages through various mechanisms [7]. Based on their role in cancer, miRNAs can be categorized into either a tumour suppressor miRNA (TS-miR) or a oncogenic miRNA (oncomiR) [8]. TS-miRs have negative influences on important cancer hallmarks, such as proliferation, invasion and avoiding apoptosis. This can be achieved by targeting oncogenes and thereby limiting their expression. In cancer, this group of miRNAs is often downregulated to achieve the opposite effect and promote tumourigenesis [8,9]. The expression of oncomiRs, on the other hand, is often upregulated in cancer. These oncomiRs commonly target tumour suppressive pathways, leading to a cell with internal stimulation towards tumour development.

Since miRNAs play an important role in the development and progression of cancer, they have been proposed as diagnostic, monitoring, prognostic as well as predictive biomarkers in this context [10]. Since miRNAs are present in liquid biopsies at a relatively stable level, they have the potential to serve as non-invasive cancer biomarkers [11]. Prior to even considering their incorporation into routine clinical care, it is important to have a thorough understanding of the physiological miRNA lifecycle, and the causes and consequences of miRNA dysregulation in cancer.

This review will go over all the steps in the physiological miRNA lifecycle, including biogenesis, target regulation, subcellular compartmentalization, secretion, uptake, stability and degradation. Moreover, cancer-related alterations in miRNA gene sequences, miRNA expression, miRNA processing, and miRNA stability will be discussed. These changes can redirect intracellular pathways to promote cancer development and progression.

## 2. miRNA Biogenesis

### 2.1. miRNA Transcription

The genes encoding for miRNAs reside at multiple genomic locations. Approximately 50 percent of miRNA genes are located intergenic, meaning that they lie in between two larger protein-coding genes [12,13]. miRNA genes can also be located intragenic, where the sequence resides within a larger host gene, usually in its intronic regions [13]. If the sequence spans the whole intron, it can be spliced out into a mirtron (discussed later on) [14]. The miRNA encoding sequence can sometimes also lie within exons, or intron-exon junctions [15]. Intragenic sequences can be positioned in the same direction as their host gene, but they can also be transcribed in the antisense direction.

Intergenic miRNAs can be transcribed using their own promoter, and/or they can be transcribed under the influence of promoters of nearby genes. One of the options is transcriptional read-throughs, where transcription continues beyond a gene’s normal termination site towards the miRNA sequence [16]. The other method involves antisense transcription, which starts at the promoter of a coding gene and generates miRNAs that are present within the upstream sequences. Intragenic genes are usually transcribed with their host gene, which allows for co-transcription and therefore co-expression [13]. However, intragenic miRNAs can also contain their own promoter [17]. Approximately 30–50% of intronic miRNAs were found to have their own promoter for RNA polymerase II (Pol II) or RNA polymerase III (Pol III) [17,18]. Most exonic miRNA genes do not contain their own promoter, and are therefore co-transcribed with their host gene [17]. Many intragenic miRNA genes are under the influence of multiple promoters, meaning that expression can be regulated in more than one way, through host gene or miRNA specific promoters.

Transcription in the canonical pathway of miRNAs starts in the nucleus and is usually performed by RNA polymerase II (Figure 1) [19]. However, a few miRNA genes that are situated downstream of repetitive sequences, such as Alu repeats, can be transcribed by Polymerase III [20]. Intronic miRNA genes are almost exclusively regulated by Pol II promoters, whilst intergenic genes can be regulated by either Pol II or Pol III promoters. After transcription, a primary miRNA (pri-miRNA) is formed that contains a stem loop, a 5′ cap and 3′ poly(A) tail. When multiple miRNA genes are located in close proximity, this cluster can be transcribed in a polycistronic fashion, forming a single, long pri-miRNA that contains multiple stem loops [21].

### 2.2. Processing of the Primary miRNA by Ribonuclease III Enzyme Drosha

After the pri-miRNA hairpin has been transcribed, it will be processed to form precursor miRNA (pre-miRNA) molecules of ~70 nucleotides long (Figure 1) [22]. This is performed in the nucleus by the microprocessor complex consisting of the ribonuclease III enzyme Drosha and its co-enzyme DiGeorge Syndrome Critical Region 8 (DGCR8). DGCR8 recognizes and binds the pri-miRNA, enabling Drosha to bind the double-stranded RNA molecule with its double-stranded RNA-binding domain (dsRBD). Subsequently, Drosha is able to cleave the pri-miRNA with its two RNase III domains (RIIIDs), which form a dimer to create a single processing centre with two catalytic sites [23]. This processing centre is able to cut the long double-stranded pri-miRNA at the stem of the hairpin structure, to excise a ~70 nucleotide long hairpin molecule, that is referred to as the pre-miRNA [22]. The cleavage leaves a 3′ overhang of 2 nucleotides at the base of the hairpin structure. This processing step can be influenced by multiple factors, including the structure of the pri-miRNA and modifications of the microprocessor proteins. Some guanine-rich pri-miRNAs can form G-quadruplexes, which can impede Drosha cleavage [24]. Moreover, both Drosha and DGCR8 can be post-translationally altered by addition of different modifications, which influence stability and localisation. Phosphorylation of Drosha is required for its localisation in the nucleus [25]. The stability of Drosha is increased by acetylation, and ubiquitination leads to degradation [26]. For DGCR8, stability is increased by phosphorylation and a higher pri-miRNA affinity can be obtained through deacetylation of the binding site [27,28]. Additionally, both proteins are also involved in a mutual regulatory loop, where they influence each other’s protein stability. DGCR8 is able to increase Drosha stability through direct protein–protein interactions [29]. Conversely, Drosha destabilises DGCR8 mRNA by excising a part of its sequence, thereby decreasing DGCR8 expression. Moreover, several auxiliary factors can increase pri-miRNA processing, such as splicing factor SRSF3 and DDX17 (p72), which both can bind a CNNC motif in the pri-miRNA [30]. The CNNC motif is conserved in a large subset of human pri-miRNAs, and binding of these factors increases processing by Drosha.

### 2.3. Nuclear Export of Precursor microRNAs Mediated by Exportin-5

After transcription and processing in the nucleus, the newly formed pre-miRNA is exported to the cytosol (Figure 1) [21]. The 3′ overhang and the base-paired 5′ end of the pre-miRNAs are recognized by Exportin-5 (XPO5) and its co-factor Ran-GTP [31]. XPO5 then binds to nucleoporin 153 (NUP153), which is part of the nuclear pore complex [32]. Next, the pre-miRNA and XPO5 are transported through the nuclear pore complex into the cytoplasm, where the GTP is hydrolysed and the pre-miRNA is released [31]. During this transport, XPO5 prevents the pre-miRNA from degradation by covering the 3′ overhang of 2 nucleotides. Several factors can influence pre-miRNA export, including the levels of Ran-GTP in the cell. Moreover, DNA damage results in the phosphorylation of NUP153 by Protein Kinase B (PKB) [32]. This phosphorylation increases the interaction between XPO5 and NUP153, which results in an increased export of pre-miRNAs. XPO5 can also be phosphorylated by the protein extracellular signal-regulated kinase (ERK), which results in a conformational trans to cis change [33]. This impairs pre-miRNA loading and therefore inhibits its nuclear export.

### 2.4. Processing of Precursor miRNA by Ribonuclease III Enzyme Dicer

After nuclear export, the pre-miRNA will be further processed by another ribonuclease III enzyme named Dicer in the cytosol (Figure 1) [21]. The 3′ nucleotide overhang and the 5′ phosphor group of the pre-miRNA are recognized by the PIWI-Argonaute-ZWILLE (PAZ) domain of Dicer [34]. After binding to the pre-miRNA, Dicer cleaves ~22 nucleotides from the 5′ end of the stem-loop to create a mature double-stranded miRNA. Dicer is able to generate mature miRNA duplexes of the right length, by placing the pre-miRNA alongside a molecular ruler [35]. Several co-factors support Dicer in this process, including transactivation-response element RNA binding protein (TRBP), adenosine deaminases acting on RNA 1 (ADAR1) and protein kinase RNA activator (PACT) [36,37,38]. The RNA helicase domain of Dicer associates with the triple dsRBD containing protein TRBP [39]. This co-factor not only aids Dicer in the binding and cleaving of its substrates, but also stabilizes Dicer [36]. The importance of both Dicer and TRBP is underlined by the observation that depletion of either protein results in loss of complex stability and decreased expression of mature miRNAs [39]. Besides TRBP, processing by Dicer is also supported by ADAR1. This protein is primarily known for its function in RNA-editing, but can also enhance Dicer function after binding to the DUF283 and DEAD-box RNA helicase domain [37]. ADAR1 has very similar functions to TRBP, and both co-factors can influence the exact site where Dicer cleaves the pre-miRNA in different ways. This leads to the production of miRNA isoforms (isomiRs) consisting of different sizes (discussed later on) [40]. Another co-factor that can bind Dicer’s RNA helicase domain, is PACT. The association of PACT with Dicer induces a preference for cleavage of pre-miRNA over other non-coding small RNA precursors [38]. PACT was also found to interact with TRBP, creating a large complex together with Dicer and AGO2 (different AGO proteins are discussed hereafter).

The processing of pre-miRNA molecules by these proteins can be influenced by the secondary structure of the RNA molecule. The presence of G-quadruplexes in the pre-miRNA usually prevents processing by Dicer, resulting in impaired miRNA maturation [41].

### 2.5. Strand Selection and miRNA Induced Silencing Complex Formation

After the ~22 nucleotide miRNA duplex has been excised from the pre-miRNA, the duplex is transferred from Dicer to the miRISC [42]. This complex primarily consists of a miRNA loaded AGO protein and the scaffolding trinucleotide repeat containing 6 (TNRC6, also called GW182) protein [43]. AGO plays an important role in strand selection and loading of the miRNA into miRISC [44]. Humans have four AGO proteins, which are all capable of binding miRNAs [42]. On average, miRNAs are loaded onto all AGO proteins with similar efficiency, although some miRNAs seem to have a bias towards one particular AGO protein. Moreover, AGO2 is the only protein that contains endonucleolytic activity, while AGO1, AGO3 and AGO4 are only capable of inducing translational repression. AGO can be recruited by TRBP, after which it can bind Dicer [39]. AGO then undergoes ATP-dependent conformational changes with the help of two chaperone proteins, Heat Shock Protein 90 (Hsp90) and Heat Shock Cognate 70 (Hsc70) [44]. After this change, AGO is ready to receive the miRNA duplex. During strand selection, one of the miRNA strands is selected and anchored into the AGO protein, while the other strand is unwound and targeted for degradation [45]. AGO can selectively bind the 5′ end of one of the strands in the binding pocket between its MID and PIWI domains [44], with a preference for the strand with less 5′ end thermodynamic stability [46]. The precise nucleotide at the 5′ end of the molecule also determines strand binding by AGO; as a result of a loop in its MID domain, Adenine (A) or Uracil (U) bases are preferred [47]. The preferential strand is called the guide strand and is in most cases the dominant strand of the duplex [45]. The 3′ end of this guide strand is localized in the PAZ domain of AGO. Once the single stranded molecule is incorporated, the miRISC is functional. The other strand, the passenger strand, is removed and degraded by RNases after passive unwinding of the miRNA duplex (Figure 1) [48]. The exact mechanism behind this degradation is not yet fully elucidated, however, one study identified the involvement of the endoribonuclease Component 3 Promoter of RISC (C3PO) [49]. After a nick in the passenger strand is induced by AGO2, C3PO can be recruited, which will aid in removing and degrading the miRNA strand.

The strand that is selected for the miRISC can sometimes change, which is called arm switching. This can be the result of altered Drosha processing, affecting the 5′ stability [50]. Moreover, uridylation by terminal uridylyl transferases (TUT) 4 and 7 can lead to altered Dicer cleavage and arm switching [40]. The arm that is selected can be tissue, species and context specific. In addition, the ratio of guide and passenger strand expression can also induce arm switching. For example, when each miRNA strand is evenly stable, both can be potentially loaded in the miRISC [51]. If the target mRNAs of, for example, the 5p arm are highly expressed, this can protect the degradation of this strand. The 5p arm is sequestered in the miRISC, protecting it from ribonucleases. This mechanism is termed target-mediated miRNA protection (TMMP) [51,52]. This changes the ratio of 5p and 3p miRNA strands, which in turn induces arm switching.

### 2.6. miRNA Isoforms

During biogenesis, different isomiRs can be formed from the same pre/pri-miRNA sequence. These can arise through specific 3′ or 5′ modifications of pri-, pre- or mature miRNAs, such as uridylation by TUT proteins as described above [40,50]. Modified pri- or pre-miRNAs can lead to altered processing by Drosha or Dicer, yielding mature miRNA sequences with variations with respect to the reference sequence. The targetome can be altered if the seed sequence is modified, which adds another layer of complexity to the miRNA target regulation [40].

### 2.7. Non-Canonical Biogenesis Pathway

miRNA biogenesis can also occur through a non-canonical pathway (Figure 1) [53]. miRNA genes that are composed of an entire intron of a host gene, mirtrons, do not follow the canonical pathway. They can be spliced from the host gene transcript directly and produce a pre-miRNA, without the need for Drosha processing. After splicing of the intron, a lariat is formed where the 5′ end is ligated to the 3′ branch point [54]. This structure is shorter compared to other pri-miRNAs, since it does not contain a lower stem, which prevents the microprocessor complex from being recruited. To further process this RNA, the circle is broken by the lariat debranching enzyme DBR1 [55]. After this, the mirtron can adopt a pre-miRNA hairpin structure and is transported into the cytoplasm via XPO5 where it enters the canonical pathway [54]. Mirtrons that contain unstructured extensions at one end, 3′ or 5′ tailed, have also been found [56]. These mirtrons are processed in the same way as untailed mirtrons, but before export to the cytoplasm, the 5′ and 3′ overhang are trimmed, for example by the 3′ to 5′ nuclear RNA exosome complex.

### 2.8. Biogenesis and Processing Rates

The biogenesis and processing as described above occurs relatively fast, compared to mRNA biogenesis, with at least 40% of mature miRNAs are produced in <5 min [57]. A study in Drosophila S2 cells (macrophage-like lineage) identified an average miRNA production rate of 228 ± 48 molecules per minute, and 4-fold lower production rate for the highest expressed miRNAs. Moreover, studies in mouse embryonic fibroblasts have determined the production rate of the most abundant miRNA, miR-21a-5p, at 110 ± 50 per cell per minute [58]. This speed is relatively high compared to the fastest reported mRNA being produced at a rate of up to 8 molecules per minute in mouse fibroblasts [59]. However, not all miRNAs are produced and processed at this speed, which is likely due to 3′ uridylation or the pre-miRNA structure [60]. Since mirtrons often contain a 3′ uridylation modification, this subgroup of miRNA are produced at a relatively lower speed. This modification inhibits processing of the mirtron by Dicer and can trigger its decay.

One major factor that influences the speed of miRNA processing is the formation of the miRISC. The loading of the miRNA into AGO happens relatively slow, typically requiring one hour [57]. This limiting factor results in the degradation of 40% of all miRNA duplexes before they can be incorporated in the miRISC complex. Moreover, transport via the XPO5 protein can be a limiting step in miRNA production [61].

## 3. miRNA Target Regulation

After the formation of the miRISC, this complex usually induces mRNA translational repression or mRNA decay [62]. If there is complete complementarity between the seven nucleotide long seed sequence and the mRNA 3′ UTR, the mRNA will be degraded. On the other hand, if target binding is imperfect as a result of central mismatches, translational repression occurs. Since in many cases the binding of a miRNA is not fully complementary, a single miRNA is able to repress a multitude of targets [63]. The complexity of miRNA target regulation is even further reinforced by the fact that there are numerous occasions where targetomes overlap, meaning that a target is regulated by multiple miRNAs. The miRISC can also induce translational activation, and is also capable of regulating transcription in the nucleus [64,65].

### 3.1. Translational Repression and Messenger RNA Degradation

The miRISC, composed of the miRNA guide strand, an AGO protein and the TNRC6 scaffolding protein, can induce both translational repression and degradation of the target mRNA [43]. Within the miRISC, TNRC6 interacts with the PIWI domain of AGO via its N-terminal Argonaute-binding domains [66]. Both proteins are guided by the guide strand to the miRNA response element (MRE), which is usually located at the 3′ UTR of the target mRNA [43]. However, binding of the miRISC to other regions of the mRNA, such as the 5′ UTR or coding sequences, can also induce translational inhibition and/or degradation. It is believed that translational repression occurs before the potential decay [67]. Translation of the target mRNA is usually repressed by interfering with the initiation or elongation steps [68,69]. After this interference, different effector proteins can be recruited that can induce deadenylation, decapping and degradation of the mRNA [62]. This process involves many effector proteins and steps that will be described below (Figure 1).

The AGO proteins do not only function in strand selection and loading of the miRNA into the miRISC, they also play an important role in translational repression and mRNA decay. When target sequence and miRNA are fully complementary, AGO2 is able to cleave the target mRNA [62]. In miRISC complexes with other AGO proteins that lack the cleaving domain, degradation requires additional proteins and complexes. If there are mismatches present between the seed sequence of the miRNA and the mRNA target, AGO is only able to induce translational repression [70]. It is capable of binding independently to the mRNA 5′ cap, which competes with translation initiation factor eIF4E binding, leading to inhibition of translation initiation [71].

Within the miRISC, TNRC6 is also an important factor for translational repression and degradation [72]. It functions as a scaffolding protein to connect the miRISC to several effectors [73]. These different effectors can lead to distinct mechanisms of translational regulation, but it is unclear if all these mechanisms are able to exert there function on their own, or act synergistically. One such mechanism is by binding to E3 ubiquitin ligase (EDD), which allows the complex to bind to silencing proteins, such as the RNA helicase p54. Additionally, TNRC6 can also competitively bind the poly(A) tail binding protein (PABP), which normally binds to eIF4G present at the 5′ cap of the mRNA to create stable circular mRNA molecules required for translation initiation [74]. The TNRC6-PABP interaction can also recruit deadenylase complexes: Carbon Catabolite Repression—Negative on TATA-less (CCR4-NOT), and Poly(A) Nuclease (PAN) 2 and 3. The platform protein TNRC6 plays an additional role in the positioning of the poly(A) tail, ensuring robust deadenylation [73]. After binding of the CCR4-NOT complex to TNRC6, PABP dissociates from the poly(A) tail, making the mRNA unstable and accessible for deadenylation [74]. The CCR4-NOT complex is the main effector of deadenylation, whilst the PAN2-PAN3 complex has a fairly limited role [73]. The latter complex can either bind to TNRC6 directly, or function after binding to PABP.

After deadenylation, the miRISC can recruit decapping complexes, such as DCP1-DCP2 [75]. TNRC6 is likely indirectly involved in the binding of these decapping factors, although the exact binding partner remains unknown [76]. The DCP1-DCP2 complex induces the release of eIF4E from the 5′ mRNA and hydrolysis of the 5′ cap. After the poly(A) tail and 5′ cap have been removed, the mRNA is unstable and subject to degradation by 5′–3′ exonucleases, such as XRN1 [75]. This whole process often occurs within processing bodies (P-bodies, discussed below).

### 3.2. Translational Activation

In addition to downregulation of target expression, there are also instances where miRNA:mRNA interaction activates translation [77]. This is dependent on multiple factors, including the miRNA binding site and availability of co-factors. Translation can be activated by binding at the conventional 3′ UTR site of the mRNA, or at the 5′ UTR. The recruitment of the co-factor TNRC6 usually leads to translational inhibition as described above, while the binding of Fragile-x-mental retardation related protein 1 (FXR1) instead of TNRC6 can induce translational activation [65]. Translation of the target can be promoted by increasing mRNA stability, blocking the binding of translation inhibitors and by recruitment of translation enhancers [77,78,79]. This miRNA-mediated gene expression upregulation is especially common in cell cycle regulation.

### 3.3. Transcriptional Regulation

The effect of miRNAs is not limited to the cytoplasm. The regulators can be transported back to the nucleus after Dicer processing in the cytosol, where they are able to control gene expression at both the post-transcriptional and transcriptional level [64] miRNA-induced transcriptional silencing or activation can be achieved in both direct and indirect ways. miRNAs can directly bind with their 5′ seed sequence to a complementary sequence in the DNA promoter, such as the TATA box [78]. After the formation of this DNA:DNA:miRNA triplex structure, the miRISC can recruit different repressive and activating effector complexes, such as histone modifiers and transcription factors [79]. The mechanism of gene expression activation by miRNAs is proposed to be similar to that of small activating RNAs [80]. miRNAs can also target transcription indirectly, by regulating expression of transcriptional regulators and epigenetic modifiers in the cytoplasm [81]. Another nuclear mechanism of transcriptional regulation, is destabilisation of the mRNA during transcription [82]. When a pre-miRNA is excised from an exon, the mRNA becomes destabilised. This way, host gene expression can be regulated by Drosha cleavage during transcription.

## 4. miRNA Subcellular Compartmentalization

miRNAs are usually present in the cytoplasm, where they induce translational repression or mRNA decay. However, miRNAs can also be located in other compartments of the cell, such as the nucleus, mitochondria and endoplasmic reticulum, or in membrane-less compartments, like granules and P-bodies [83,84,85,86,87]. The localisation of miRNAs influences their function.

### 4.1. Membrane Compartments

#### 4.1.1. Nucleus

As explained previously, nuclear miRNAs can regulate transcription. For this to be possible, the miRISC needs to be transported into the nucleus, which is performed by importin proteins and their co-factor Ran-GTP [81]. TNRC6 of the miRISC contains a nuclear localisation signal (NLS) that can possibly be bound by a nuclear importer protein, however, no specific binding interaction has been established yet. Transport into the nucleus can also occur through association of Importin-8 (IPO8) with AGO2 [82]. Most miRNA sequences do not contain a NLS and can therefore only be exported and imported outside and into the nucleus bound to miRISC. However, one study identified a mature miRNA, miR-29b, harbouring a hexanucleotide NLS sequence at its 3′ end [83].

#### 4.1.2. Mitochondria

miRNAs have also been reported to be present in mitochondria from several species [84]. This organelle contains its own mitochondrial DNA (mtDNA), which includes miRNA gene sequences. However, since processing proteins like Drosha are not present in the mitochondria, the biogenesis and function of these specific miRNAs is not well understood. Mature genomic DNA-derived miRNAs can also localize and function in the mitochondria. The N-terminus of AGO2 contains a mitochondria localisation sequence and can thus be transported into the mitochondria together with the mature miRNAs [85]. Polynucleotide phosphorylase, a protein found in the inner membrane space of the mitochondrion, is also believed to play a role in the import of pre-miRNAs to mitochondria [86]. This protein transports specific RNAs with a stem-loop structure and an overhang at the 3′ site, but degrades other RNAs.

After a miRNA has been transported into the mitochondria, it can for instance play a role in the intrinsic apoptotic pathway [87]. miR-15a and miR-16-1, for example, can induce the release of cytochrome c. Furthermore, miR-181c can be imported into the mitochondria and inhibit mitochondrial encoded cytochrome c oxidase subunit 1, thereby compromising the electron transfer chain [88]. This data indicates that miRNAs, regardless of whether they are encoded by mtDNA or have been imported from the cytoplasm, can have a regulatory function on mitochondrial processes.

#### 4.1.3. Endoplasmic Reticulum

miRNAs have a function in the endoplasmic reticulum (ER) as well, where they promote RNA cleavage, protein modification and the ER stress response [84]. miRNAs are mainly present on the rough ER, which is enriched with membrane-attached ribosomes that are necessary for protein translation. Localization of miRNAs to the ER membrane is not yet fully elucidated, however, it likely involves TRBP and/or PACT [85]. During ER stress, miRNAs can influence protein folding and modification, although the mechanism is poorly understood. Moreover, miRNAs are involved in a regulatory feedback loop during ER stress. Cellular stress can induce upregulation of miR-26a that targets the core ER stress effector eIF2 [89]. Downregulation of this effector through miRNA translational regulation reduces ER stress.

### 4.2. Membrane-Less Compartments

Membrane-less compartments are formed through phase separation [84]. These compartments play a crucial role in maintaining homeostasis, especially during cellular stress response. This requires massive and rapid changes in gene expression.

#### 4.2.1. Stress Granules

Stress-granules are cytoplasmic non-membranous cellular compartments that function in mRNA translation, cellular stress and homeostasis [84]. They primarily contain RNA sequences and RNA binding proteins, such as Dicer and AGO. Upon cellular stress, the interaction between Dicer, Hsp90 and AGO increases within stress-granules, which leads to a decreased catalytic activity of Dicer [90]. Thus, stress induces the formation of stress-granules, in which the binding of Dicer with its co-factors is altered, leading to reduced miRNA biogenesis.

#### 4.2.2. Processing Bodies

Another important non-membranous cellular compartment where miRNAs reside in are P-bodies [91]. These are distinct and dynamic cytoplasmic foci with key roles in the regulation of cellular mRNA fates. P-bodies are compartments where mRNAs are directed to, either to be degraded or to be stored for subsequent return to translation. The RNA-binding protein PatL1, involved in deadenylation-dependent decapping of mRNAs, plays an important role in their formation [92]. The majority of the proteins present in P-bodies are involved in translational repression or RNA decay. These proteins include, but are not limited to, TNRC6, AGO, decapping enzymes and RNA helicases. The maintenance of P-bodies depends on several proteins, for example, ubiquitination of DCP1 leads to the loss of P-bodies. However, especially the presence and density of translationally repressed mRNAs are important in P-body maintenance. Repressed mRNAs can be targeted to a P-body by TNRC6 [93]. Studies have suggested that repressed mRNAs may return from P-bodies to the cytosol to re-enter translation [94].

## 5. miRNA Secretion and Uptake

The expression of miRNAs is different among tissues and biofluids [95], which can be explained by tissue-specific promoters [17] and the selective release of miRNAs [96]. Aside from passive secretion/leakage through necrosis, miRNAs can be actively exported out of the cell through release of extracellular vesicles, such as exosomes, microvesicles (MVs) or apoptotic bodies [97]. These groups are categorized by size and origin. All three are very important in cell–cell communication, in physiological and pathological conditions [98]. Moreover, it is thought that miRNAs are actively secreted to maintain cell homeostasis [99].

### 5.1. Exosomes

Exosomes (<100 nm) are endosome derived membrane MVs that contain a specific set of proteins, lipids and nucleic acids, including miRNAs [97]. The composition of molecules within an exosome is highly heterogeneous, depending on the cell type and pathophysiological condition. The miRNA expression levels in exosomes can be different compared to their donor cells [100]. This implies that the release of miRNAs in exosomes is an active process, that requires ATP [96]. The biogenesis of exosomes starts at the cell membrane where an early endosome is formed through inward budding [97]. During maturation, a multi vesicular body (MVB) is formed that contains multiple intraluminal vesicles through invagination of the endosomal membrane. These smaller vesicles can be released in the extracellular space and are called exosomes.

During this process, specific molecules are incorporated into the vesicles, including miRNAs. Multiple different proteins are involved in this, but the exact mechanism is not yet fully understood. Sorting of specific molecules into exosomes can be performed by the Endosomal Sorting Complex Required for Transport (ESCRT) or can be done independently of this complex [101]. Guiding of miRNAs into exosomes is believed to occur via an ESCRT-independent mechanism involving miRISC, SUMOylated heterogeneous nuclear ribonucleoproteins (hnRNPs), or membrane proteins. The first group of proteins that is involved in miRNA sorting into exosomes is the miRISC, specifically the AGO protein [102]. Here, AGO levels and phosphorylation play an important regulatory role. However, 90% of AGO bound miRNAs stay in the cytosol [103]. A second group of proteins involved in exosomal miRNA sorting is the hnRNP family. The SUMOylated hnRNPA2B1 protein can for instance recognize a specific 3′ EXO (GGAG) motif in miRNAs, and load them into exosomes [104]. Two other hnRNP family members might also be involved in miRNA sorting, namely hnRNPA1 and hnRNPC [103]. Moreover, membrane proteins, such as nSMase2 and Vps4A are also involved in sorting miRNAs into exosomes [105].

miRNAs can enter exosomes in a protein independent way as well. This mechanism relies on the interaction between a miRNA sequence and the membrane of a MVB [103]. During the formation of the exosomes through budding, the miRNAs that are retained to the membrane will become located in the intraluminal vesicles/exosomes. Moreover, post-transcriptional modifications can influence miRNA sorting, specifically 3′ end uridylation, but the underlying mechanism is not yet fully understood [97].

The exact process of exosome secretion has not been fully established. One option is that MVBs interact with the plasma membrane, after which the content is released [101]. This depends on interactions with multiple factors, including microtubule and microfilament cytoskeleton, molecular motors, Rab GTPases and *N*-ethylmaleimide-sensitive factor attachment protein receptor (SNARE) proteins. MVBs are transported to the plasma membrane using molecular motors that walk along the cytoskeleton [97]. Rab27 regulates this transport and the fusion of the vesicles, while Rab35 controls the docking with the plasma membrane. SNAREs are necessary for fusion with the plasma membrane. Ca^2+^ is capable of stimulating this process and the release of the exosomes [103].

The formation of MVBs is not only required for miRNA secretion, but these compartments also influence miRNA function. MVBs play a role in translational repression, since they promote recycling of the miRISC to the cytosol, leading to enhanced miRNA function [106]. Additionally, the endosomal trafficking pathway plays a role in distribution of miRNAs throughout the cell. This effect has been identified in neurons, where miRNAs where transported to the distal axons via lysosomes/late endosomes, where they could be processed [107].

Not much is known about the uptake mechanisms of extracellular miRNAs by target cells. In the case of exosomal miRNAs, it is presumable that the vesicle attaches to the target cell via tetraspanins, since these membrane proteins are highly expressed in exosomes [108]. They contribute to the selective uptake of exosomes by target cells. After attachment, the exosome can fuse with the plasma membrane or enter the cell via phagocytosis, macropinocytosis or receptor-mediated endocytosis [109].

### 5.2. Microvesicles

MVs (<1000 nm) are a heterogeneous population of membrane vesicles that are formed by outward budding of the plasma membrane [97]. They are commonly formed in platelets, endothelial cells and red blood cells and come in variable shapes [110]. The formation and release of MVs is a complex process that involves cytoskeleton reorganization and loss of the physiological asymmetry of the membrane bilayer. They are mainly formed in regions of the plasma membrane that contain lipid-rich microdomains [111]. The underlying mechanism of this process is not fully understood, but Ca^2+^ plays a crucial role in the formation and release of MVs. Ca^2+^ is capable of inducing cytoskeleton remodelling and altered lipid regulation, and by doing so, MVs can be formed and released.

Precursor and mature miRNAs can be packaged into MVs, and are transported between cells or secreted into biofluids. When associated with the miRISC, these miRNAs can regulate translation in recipient cells [111]. Moreover, single stranded passenger strands have also been identified in MVs that were not bound by the miRISC. These strands are protected against degradation by RNAses within the MVs, however, it is not known if they have a biological function, or are simply degraded after entering a neighbouring cell. The mechanism of miRNA sorting into the MVs has not been elucidated yet.

### 5.3. Apoptotic Bodies

Extracellular miRNAs can originate from apoptotic or necrotic cells [112]. Apoptotic bodies (>1000 nm) are formed during late stages of apoptosis through blebbing of the apoptotic cell membrane [97]. These contain several intracellular fragments, including cellular organelles, membranes, and cytosolic contents, like miRNAs [110]. The miRNAs present in these apoptotic bodies can serve as paracrine signalling mediators [98]. After release, apoptotic bodies can be engulfed by phagocytes, or they can enter neighbouring cells, which can possibly influence the transcriptional and translational regulation within this cell [98,110]. One study identified that release of miR-126 in apoptotic bodies can alter the chemokine response in neighbouring cells [113]. However, this is one of the few studies that have identified a specific effect of miRNA secretion in apoptotic bodies.

## 6. miRNA Stability

Stability is an important factor in a miRNA’s lifecycle. The overall high stability of miRNAs is necessary for their accumulation in the cell, which is required since relatively high concentrations are needed to perform their function [57]. The half-life of miRNAs can vary from hours [114] to weeks [115]. Around 40% of the mature miRNAs duplexes are degraded before they can be incorporated into miRISC [57]. The average half-life of a guide strand is 11.41 h, while the average half-life of the passenger strand is no more than 0.69 h [57]. Regardless, this is still relatively long compared to the 4.8 min average half-life of a mRNA [116]. In contrast to other miRNAs, it seems that miRNAs involved in the cell cycle are relatively unstable [114]. This is necessary to guarantee the rapid changes in expression levels throughout the different cell cycle phases. The steady-state levels of mature miRNAs seem highly correlated with their biogenesis rate, especially with their transcription rate. However, other factors also influence miRNA levels, including the CG content, binding proteins, mRNA:miRNA interactions, miRNA editing, miRNA methylation, binding to competing endogenous RNAs (ceRNAs) and miRNA:miRNA interactions [37,117,118,119,120,121,122,123]. Moreover, miRNAs in biofluids, such as serum and urine, are also relatively stable and resistant to degradation by ribonucleases, since they are secreted into MVs, exosomes, apoptotic bodies or bound to proteins, such as AGO or HDL [111,119,120].

### 6.1. Intrinsic Stability

The sequence of a miRNA can influence its stability. For example, the amount of GC or AU bases influences the thermodynamic stability of a miRNA [117]. GC-enriched miRNAs seem to be more stable than AU-rich miRNAs. It is tempting to think that this has something to do with the Watson–Crick base pairing, however, this seems unlikely since the mature miRNA interacts with binding proteins in a sequence-independent manner. Moreover, some miRNAs contain specific sequences that can make them more susceptible for decay [124]. For example, miR-382 contains the 7 nucleotides GGAUUCG at the 3′ end, making this miRNA more susceptible for degradation. Mutations within this sequence were shown to increase miRNA stability. Moreover, uridine-rich elements can also increase susceptibility of miRNA decay [124]. The exact mechanisms and proteins involved in the intrinsic stability of miRNAs remain to be elucidated.

### 6.2. Binding Proteins

miRNA turn-over rates highly depend on their interaction with proteins. AGO and HDL proteins increase miRNA stability by protecting them against RNases [118,119]. Stability depends on which AGO protein is bound, since AGO2-loaded miRNAs have a longer half-life than miRNAs loaded into other AGO proteins [57]. Specific interactions between proteins and a (subset of) miRNA that affect stability have also been identified. For example, a member of the signalling transduction and activation of RNA family, Quaking, can bind directly to mature miR-20a, thereby increasing its stability [125].

### 6.3. Target-Directed miRNA Degradation or Protection

One phenomenon that can decrease miRNA stability is target-directed miRNA degradation (TDMD). During this process, the interaction of a miRNA with its target mRNA can cause a change in miRNA stability [120]. Within the miRISC, the miRNA is usually highly stable because its ends are protected by AGO [126]. When a miRNA binds to its target with high 3′ end complementarity, TDMD can induce dissociation of the miRNA from AGO. As the 3′ end of the miRNA becomes exposed, it is available for miRNA tailing. In this process, the miRNA is tagged with uridyl or adenyl tags [127]. Uridylation marks are placed, for example, by TUT7, which marks the molecule for degradation by the 3′ to 5′ exoribonuclease DIS3-like exonuclease 2 (DIS3L2) [128]. Moreover, an adenylated 3′ tail can also lead to miRNA degradation by DIS3L2 [129]. A possible explanation for TDMD, could be that the complementarity changes the conformation of AGO. This could result in the displacement of the 3′UTR from the PAZ domain of AGO, resulting in 3′ modifications and therefore enhanced exoribonuclease activity [126]. miRNA degradation by TDMD is also possible through AGO modifications that destabilize the miRNA. Zinc Finger SWIM type 8 is an ubiquitin ligase that can recognize the free PAZ domain after mRNA:miRNA binding and ubiquitinate AGO [130]. This marks AGO for destruction in the proteasome. Degradation of this protein makes the miRNA unstable and results in its degradation.

In contrast to the above, some studies indicate that adenylation increases miRNA stability, suggesting that the effect of such modifications can vary [131]. Moreover, interaction between the miRNA and the target mRNA can also induce the opposite effect of TDMD, namely TMMP [52]. This process, as explained in the section “miRNA biogenesis”, can inhibit miRNA degradation. One study showed that a decrease in complementarity through mutations of the seed sequence led to increased degradation of the miRNA [52]. However, not much is known yet about the mechanism of TMMP.

### 6.4. Adenosine to Inosine Transition

Adenosine to inosine transition is a modification that can affect miRNA biogenesis and decrease miRNA stability [37]. Which miRNAs are edited, and in what way, is called the editing pattern [132]. Editing changes the targetome of the miRNA, and since the pattern is tissue specific, it can be used to achieve different cell fates [133]. Adenosine to inosine modification is catalysed by ADAR1 or 2. ADAR1 is predominantly located in the cytoplasm, but can also be located in the nucleus, while ADAR2 is only present in the nucleus [37]. Since inosines are recognized as a guanosine, this modification within the seed sequence will influence mRNA binding [134]. Furthermore, ADAR modifications can affect miRNA biogenesis, for example pri-miRNA processing [132]. In this case, processing of Drosha can be inhibited through the presence of an inosine in the pri-miRNA. Similarly, ADAR1 editing of the pre-miRNA can alter cleavage by Dicer [135]. ADAR is not only responsible for adenosine to inosine transition, but it can also interfere with miRNA processing without base editing [136]. A catalytically inactive ADAR protein is capable of binding pri-miRNA structures, which inhibits binding and cleavage by Drosha. This indicates that the mere binding of ADAR can influence miRNA processing, independently of nucleotide editing. Moreover, ADAR can also regulate the activity of processing factors like Drosha, Dicer and DGCR8 [37]. If ADAR1, for example, binds to the DEAD-box RNA helicase domain of Dicer, its activity is enhanced, therefore resulting in increased processing of pre-miRNA and promoting miRNA loading into miRISC.

The adenosine to inosine modifications can also be recognized by the Tudor-SN (TSN) nuclease that degrades adenosine to inosine edited double-stranded RNAs by a process called TSN-mediated decay (TumiD) [137]. TSN is a part of miRISC and functions in degradation of adenosine to inosine containing pri-, pre-, and mature RNAs [132]. Since TSN is located in the cytoplasm, degradation of pri-miRNAs requires nonspecific export of these molecules out of the nucleus.

### 6.5. miRNA Methylation

Another modification that can alter mature miRNA stability is methylation. This modification is reversible and performed by methyltransferase-like (METTL) 3 and 14 for methylation on an adenine, METTL1 for methylation on a guanine, and NOP2/Sun RNA methyltransferase family member (NSUN) 2 for methylation on a cytosine [121]. Methylation impairs a miRNA’s ability to downregulate its targets. Methylation of mature miRNAs causes large structural changes, which affect AGO binding. Moreover, these changes can also affect the seed sequence, thereby modifying binding of the miRNA to its target.

### 6.6. Competing Endogenous RNAs

ceRNAs can compete with the target mRNAs for binding to the miRNAs [122]. All kinds of RNAs with a MRE have the potential to be a ceRNA. Especially long non-coding RNAs (lncRNAs) function as a miRNA sponge to capture miRNAs and therefore repress their function [138]. lncRNA are RNA transcripts from 200 nucleotides to over 100 kilobases in length [139]. Their miRNA binding sites allow them to function as decoy mRNA molecules, to clear miRNAs from the cell [138]. A similar principle is at play for circular RNA (circRNA):miRNA interaction and regulation. As the name already states, circRNA are circular fragments of RNA that are highly stable because of their shape [140]. The circRNA expression is often tissue and timing specific and highly conserved across species. Both lncRNA and circRNA have been termed miRNA sponges, for their role in depleting miRNA molecule availability [138,140].

In line with the competitive function of lncRNA and circRNA, transfer RNAs (tRNA) can also be viewed as miRNA competitors [141]. These molecules are originally designed to deliver an amino acid to the ribosome for protein synthesis [142]. However, when they are misfolded, they can become substrate for RNases and be processed into small molecules. There are two types of small ncRNAs that can derive from tRNAs; tRNA-derived small RNA (tsRNA) and functional miRNAs. The tsRNAs are produced after the tRNA is cleaved by RNaseZ [141], whilst miRNAs are produced following the mirtron pathway, through Dicer [142]. The tsRNAs are known to compete with endogenous miRNAs in the RNA interference machinery, thereby affecting miRNA stability and altering cellular regulation [142]. These molecules can in part explain why in many studies there would still be a baseline miRNA expression, even when crucial elements of the biogenesis machinery were knocked down.

Another interaction affecting miRNA stability is miRNA:miRNA interaction [123]. Based on Watson–Crick base pairing models, it was found that some mature miRNAs can bind to other mature miRNA molecules with more affinity than they bind to their own passenger strand. Lai et al. predicted in 2004 that the interaction between miR-5 and miR-6 was stronger than the interaction of both miRNAs with their own complementary strand in the miRNA duplex [123]. Although it is still unclear how miRISC-bound miRNAs can bind to each other, and how this results in miRNA regulation, this miRNA:miRNA pairing can either lead to higher miRNA stability, by preventing degradation, or interfere with target regulation. In addition, interactions between mature miRNAs and pre-miRNAs have been described, resulting in auto- or heteroregulatory positive and negative cellular feedback loops [143,144]. Besides direct interaction, miRNAs are also able to affect the expression of other miRNAs indirectly, by targeting transcription factors or the biogenesis machinery [145,146].

## 7. miRNA Dysregulation in Cancer

Several factors can directly affect miRNA expression and function in cancer. These include single nucleotide polymorphisms (SNPs) in miRNA genes, SNPs in promoter elements, epigenetic changes to the miRNA gene, and chromosomal changes [147,148,149,150]. miRNA expression and function can also be dysregulated by indirect factors, including changes in host gene expression, SNPs in target genes, alterations in transcription factors, altered miRNA stability, modified miRNA editing, altered miRNA processing, and interaction with non-coding RNAs [151,152,153,154,155,156,157,158]. These alterations often lead to an altered targetome, which suggest big changes in mRNA expression levels in the cell [132]. Here, we review current knowledge on the mechanisms of miRNA dysregulation, and provide appropriate examples within the cancer field. A list of dysregulated miRNAs mentioned in this review is provided in Table 1.

### 7.1. Single Nucleotide Polymorphisms in miRNA Genes

One essential element of miRNA-mediated target regulation, is complementarity between miRNA and target mRNA. However, when a miRNA gene has acquired SNPs in the seed sequence, this interaction can be affected (Figure 2A) [147]. Because miRNA molecules are small, the accumulation of loss-of-function or gain-of-function point mutations in their genes are rare events, especially in the smaller, more conserved seed region. Even though the accumulation of SNPs in mature miRNAs is rare, they are a key cause for miRNA dysregulation in cancer. One example of a SNP in the seed region associated with increased cancer risk is the rs3746444 SNP in *mir-499a-3p* [147]. This SNP is able to directly influence target recognition, resulting in a gain of new targets, as well as loss of a substantial number of original targets. The gained targets are concentrated around cancer related pathways, such as the NF-κB and PI3K-Akt pathway, leading to increased risk of developing gliomas [159].

In addition to binding efficiency, SNPs can also affect mature miRNA levels via impaired miRNA processing at the pri-miRNA, or pre-miRNA level through altered binding by Drosha or Dicer (Figure 2A) [160]. For example, SNP rs7372209 in the *mir-26a-1* gene affects Drosha processing of the pri-miRNA molecule, resulting in lower mature miRNA expression. This oncogenic SNP was first detected in bladder cancer [160], and later in oral squamous cell carcinoma, oesophageal squamous cell carcinoma, lung cancer, cervical cancer, breast cancer, and colorectal cancer [161]. Interestingly, SNPs in miRNA genes can also enhance processing, resulting in more mature miRNAs. This is for example observed for *mir-196a-3p*, where SNP rs11614913 promotes processing from pre-mir-196a to mature miRNA [162]. The higher levels of mature miRNAs were associated with decreased survival in non-small cell lung cancer (NSCLC).

### 7.2. Single Nucleotide Polymorphisms in miRNA Promoter Elements and Transcription Factor Binding Sites

SNPs in the promoter region of the miRNA gene can result in stronger or weaker binding by transcription factors (TFs) and other components of the transcription machinery (Figure 2B). In >50% of human cancers, the MYC transcription factor is involved, since this family of TFs control the expression of around 15% of the entire genome [188]. One of the genes under the control of c-MYC, is *mir-378* [189]. This miRNA inhibits TOB2, a tumour suppressor gene, and thereby activates transcription of cyclin D1. Roughly in the same region as the TF binding site of c-MYC in the *mir-378* gene, SNP rs2340620 is located [148]. This SNP decreases c-MYC binding and therefore lowers miR-378 expression in breast cancer, leading to a better prognosis.

### 7.3. Epigenetic Changes to miRNA Genes in Cancer

Promoter regions of miRNA genes can also be regulated by epigenetic modifications (Figure 2C). This can be in the form of suppressive CpG island hypermethylation or transcriptional activation/repression through histone modifications [149,190,191]. These regulatory mechanisms can be exploited by tumours, to increase oncomiR and decrease TS-miR gene expression. For instance, in prostate cancer, miR-130a expression correlates to the methylation status of its promoter [163]. miR-130a normally reduces cell viability and invasion capacity via regulation of its targets. However, when the gene’s promoter is hypermethylated, less miRNA is produced, resulting in impaired inhibition of these cancer hallmarks.

### 7.4. miRNA Dysregulation through Host Genes

Whereas intergenic miRNA genes have their own promoters for transcription by Pol II or Pol III [16], intragenic miRNA expression can in some occasions be regulated by the host gene [13]. Altered expression of the host gene, by for example hypermethylation, can decrease miRNA levels (Figure 2D). The strongest correlation between miRNA and host gene expression is found for mirtrons [53]. For instance, the 5′-tailed mirtron miR-4728 is encoded by an intron of the HER2 gene [158]. HER2 transcription produces miR-4728 to the point that expression of miR-4728 has been suggested to accurately mark HER2 status in HER2-positive breast and gastric cancer. Among other functions, miR-4728-3p has been shown to regulate oestrogen receptor alpha (ERα) expression, to act as a negative feedback mechanism for HER2 signalling by modulating the MAPK pathway, and to stabilise oncogenic miR-21-5p through inhibition of the non-canonical poly(A) polymerase PAPD5 involved in non-templated 3′ miRNA adenylation [158].

Copy number variations (CNVs) can also influence miRNA expression (Figure 2E). When a part of the chromosome is amplified or deleted, the level of gene expression is increased or decreased, respectively (Figure 2E). The presence of a miRNA gene within one of these regions could increase cancer risk or influence tumour properties. As it turns out, around 50% of miRNA genes are located at these fragile sites of the genome [150]. What regions are frequently deleted or amplified, contain breakpoints, or are known to translocate, is often tumour type specific. For example, the cluster of *mir-15* and *mir-16* genes is located in a fragile site [164]. The genomic region of this cluster is deleted in more than half of B-cell chronic lymphocytic leukaemia (B-CLL) cases [164], as well as 60% of high grade prostate cancer cases [165]. The miRNAs in this cluster have tumour suppressive properties, by targeting the oncogene BCL2 [166]. This interaction between miRNA and mRNA is a key regulatory mechanism of BCL2 expression. Consequently, when this cluster is deleted by a CNV, a vital component of BCL2 control is absent, promoting tumourigenesis. When miR-15 and miR-16 expression is restored, this process is countered and apoptosis induced.

### 7.5. miRNA Dysregulation through Transcription Factors

TFs that control expression of both a miRNA gene and the target of that miRNA, allow for a closer regulation. The TF itself regulates target gene expression in a global manner, whereas TF-induced or repressed miRNAs are able to fine-tune target expression [192]. TFs involved in known tumour suppressor or oncogenic pathways, such as TP53 and MYC, respectively, are often affected in cancer. Dysregulation of these TFs can also result in miRNA dysregulation, which could enhance tumourigenesis [151]. However, since miRNAs have such a broad range of targets, the oncogenic effect of disturbances in TF expression could be countered. For example, some miRNAs can create negative feedback loops with their own overactive transcription factors (Figure 3A). Such bidirectional control has for instance been described between MYC and the *mir-17-92* cluster [193]. Another miRNA gene that is regulated by MYC is *mir-29* [167]. MYC is upregulated in bladder cancer, which represses miR-29 expression, leading to increased levels of the target DNA methyltransferase 3 alpha (DNMT3A). This protein can in turn silence phosphatase and tensin homolog (PTEN), which promotes urothelial tumourigenesis.

### 7.6. Factors Affecting miRNA Processing in Cancer

Altered miRNA processing is another mechanism of miRNA dysregulation in cancer. Different steps of the biogenesis process can be disturbed. Processing by Drosha can be affected in several ways; mutations in the Drosha gene, miRNA-induced regulation of Drosha expression, post-translational modifications (PTM) of Drosha and availability of co-factors (Figure 3B) [155,194,195]. As discussed earlier, SNPs in specific miRNA genes can affect processing of pri-miRNAs and reduce or increase mature miRNA expression of one particular miRNA. On the other hand, when the Drosha gene contains SNPs or mutations, a larger set of miRNAs can be affected at a time. In Drosha, a small number of SNPs/mutations have been observed [196]. More specifically, missense mutations in the second RIIIB have been identified [155]. These missense mutations, including the hotspot E1147K, impair miRNA expression of multiple TS-miRs at once, even more than a null mutation would, by acting in a dominant-negative way. This means that the mutation not only prevents processing by Drosha, but also lowers pri-miRNA expression levels, leading to very low mature miRNA levels. Rakheja et al. identified a new subtype of Wilms tumours driven by such mutations, leading to widespread dysregulation of miRNA biogenesis. The Drosha mutant was shown to be completely paralysed in processing pri-miRNAs, resulting in lower expression of both 5p and 3p strands of essential TS-miRs, such as the entire let-7 miRNA family. The loss of miRNA expression is, however, never complete as a result of alternative miRNA processing pathways, which is in line with the hypothesis that Wilms tumours need some background miRNA activity to be able to progress.

Another way Drosha processing can be altered, is by post-transcriptional control. Drosha mRNA contains binding sites for several miRNAs in its 3′ UTR [146]. One such Drosha targeting miRNA is the tumour suppressive miR-27b. In bladder cancer, the rs10719 SNP in the Drosha mRNA’s 3′ UTR impairs miR-27b binding, resulting in increased Drosha levels. Overexpression of Drosha has been associated with increased proliferation and evasion of apoptosis. The co-factor of Drosha, DGCR8, can also be affected in cancer. In several cancers, such as colorectal cancer [195] and ovarian cancer [180], upregulation of DGCR8 was found, suggesting an oncogenic role for this protein.

Processing of miRNAs can also be altered at the pre-miRNA level (Figure 3C). Similar to the mutation found in RIIIB of Drosha, the RIIID segment in Dicer is also a hotspot for mutations in non-epithelial ovarian cancer [197] and Wilms tumours [155]. However, where mutation of RIIIB in Drosha resulted in full paralysis of the enzyme, in Dicer, the inactivation of RIIIB only affects 5p strand processing. Targets that were originally suppressed by the 5p arm are therefore overexpressed and the resulting 3p strand bias was shown to have oncogenic effects, depending on cell type and developmental stages [198]. Additionally, several SNPs can affect Dicer expression and have profound effects on the production of mature miRNAs. One example of a SNP that decreases Dicer expression, is SNP rs3742330 in the 3′ UTR of the Dicer mRNA [168]. The SNP decreases Dicer mRNA stability and thereby protein expression. This is due to the creation of new target sites for miR-3622a-5p and miR-5582-5p. The presence of this SNP in Dicer has been linked to higher overall survival (OS) in mature T-cell lymphoma (TCL) [169] and hepatocellular carcinoma [170]. SNPs in Dicer co-factors are also found to influence miRNA processing, not directly via Dicer cleavage, but indirectly through miRISC loading [160]. Moreover, Dicer and its co-factors can contain PTMs, such as phosphor groups, that influence their function and location in the cell [49,199]. Phosphorylation of Dicer by ERK induces translocation to the nucleus, leading to altered miRNA processing in the cytosol.

### 7.7. Factors Affecting Precursor microRNA Export in Cancer

Dysregulation of pre-miRNA export has also been identified in different cancer types. In cancer, XPO5 up- or downregulation can be the result of alterations at both the gene and protein level [156,200]. One way XPO5 expression can be altered, is by inactivating gene mutations (Figure 4A) [156]. Premature termination codons and frameshifts produce truncated proteins that can no longer bind pre-miRNAs. When full-length XPO5 expression is restored, proliferation is inhibited, suggesting a tumour suppressive role for XPO5 in colorectal cancer. Epigenetic modification of the XPO5 gene can also play a role in regulating its protein levels [201]. In breast cancer, the promoter and first exon of the XPO5 gene are hypomethylated, which explains the increased expression. Another way XPO5 expression is controlled, is through miRNA-induced post-transcriptional regulation [202]. This phenomenon was already reported for Drosha, Dicer and AGO regulation, and also plays a role here. Overexpression of TS-miR-138 reduces XPO5 expression by downregulating a protein that is important for XPO5 stability. When this miRNA is downregulated, XPO5 becomes more stable which promotes tumourigenesis. Protein levels of XPO5 can also be affected via phosphorylation by ERK [33]. Usually, XPO5 is found in both the cytoplasm and nucleus, however, after phosphorylation, an additional protein can bind nuclear XPO5 and keep the protein in the nuclei of hepatocellular carcinoma cells [203]. XPO5 mediated translocation of pre-miRNA from the nucleus to the cytoplasm is therefore reduced in hepatocellular carcinoma, leading to less mature miRNA availability in the cytoplasm. Furthermore, XPO5 dephosphorylation favoured the distribution of XPO5 into the cytoplasm, leading to hepatocellular carcinoma inhibition in vitro and in vivo [204].

### 7.8. Factors Affecting Strand Selection in Cancer

Dysregulation in cancer might also change the strand that is incorporated in the miRISC, to be favouring tumourigenesis. This phenomenon is seen in many cancer types, for example in BRCA-deficient breast cancer cells, where miR-223-5p levels were shown to be enriched and miR-223-3p levels decreased [171]. This shift from the dominant miR-223-3p arm to the passenger strand, is a result of failing homologous recombination. This puts more pressure on DNA repair via the non-homologous end-joining (NHEJ) machinery, which is normally repressed by miR-223-3p. The BRCA-deficient cells compensate for this shift to NHEJ, by switching to the miR-223-5p arm that does not repress the NHEJ machinery. Moreover, a prospective role of TMMP in cancer specific arm switching has been proposed (Figure 4B) [172]. For instance, the miR-193a-5p/miR-193a-3p ratio was shown to be decreased in breast cancer tissue compared to adjacent normal tissue, due to an upregulation of miR-193a-3p target genes.

### 7.9. miRNA Dysregulation through miRNA-Induced Silencing Complex Assembly

As the main component of the miRISC, there are several aspects that can affect the pivotal role of AGO in this complex (Figure 4C). First of all, transcription of the AGO genes can be altered leading to decreased miRNA functionality. For example, frameshift mutations in AGO2 leading to a loss of expression are common in gastric and colorectal cancers [157]. Moreover, as a result of copy-number variations, altered expression of AGO proteins was seen in breast cancer, ovarian cancer, and melanoma [205]. In contrast, AGO2 expression was found to be higher when SNP rs3928672 was present in nasopharyngeal cancers, and the SNP was associated with a significantly increased risk of the disease and with tumour promoting properties [206]. Furthermore, AGO mRNAs are subject to post-transcriptional regulation by miRNAs [207]. The miR-132 molecule was found to specifically suppress levels of AGO2, which also altered expression of miR-221 and miR-146a. After miR-132 activation, these miRNAs were down- and upregulated, respectively. Although the mechanism that induces the different responses to loss of AGO2 is unclear, this phenomenon does suggest that there are other mechanisms at play that influence miRNA functionality. Additionally, the Hsc70/Hsp90 chaperone machinery is frequently overexpressed in tumours, to ensure normal protein folding and enhance cell survival. These proteins also allow cells to ignore tumour suppressive signals and even make them oncogenic [44]. For the miRNA-induced silencing machinery, this could mean that AGO resides in a more stable conformation that induces miRISC activity. Finally, transfer of miRNAs to the AGO proteins can also be affected. This process is highly dependent on AGO’s conformation and interaction with Dicer. Within a tumour, cells in the middle can reside in hypoxic conditions. Under these circumstances, epidermal growth factor receptor (EGFR) acts as an oncogene and is able to phosphorylate AGO2 at Tyr393 [208]. This leads to a weaker interaction with Dicer and interferes with the loading of specific miRNAs that contain large loop structures. Most of these miRNAs have tumour suppressive functions in healthy cells, making the phosphorylation of AGO2 tumour promoting. Tyr393-AGO2 thereby enhances cell survival and invasiveness under hypoxia and correlates with poorer OS in breast cancer patients.

In addition to the mechanisms affecting AGO in cancer, its co-factors in the miRISC can also be affected. For example, frameshift mutations in TNRC6 leading to total loss of expression were found in gastric and colorectal cancer [157]. Although overexpression of the miRISC co-factor p54 has been observed in several malignancies [209], no reports have yet linked p54 upregulation in cancer to miRISC dysregulation.

### 7.10. miRNA Dysregulation through Target Genes

In addition to genetic alterations in the miRNA gene, the mRNA target sequence can also be altered. SNPs in target sites influence the regulatory properties of miRNAs, by creating new targets and loss of original targets (Figure 5A). For example, in almost 4000 breast cancer cases in Europe, SNP rs1982073 (29T>C) was found in the coding sequence of the TGFB1 gene, which adds a second target site for miR-187 [152]. The extra recognition site in the mRNA results in significantly more suppression of the target. In line with TGF-β acting as a tumour suppressor during the early stages of cancer development, a meta-analysis of 39 case–control studies indicated that rs1982073 was significantly associated with breast cancer risk in the Asian population [173]. In addition to SNPs affecting miRNA:mRNA target recognition, SNPs can also influence the degree of silencing.

### 7.11. Factors Affecting miRNA Degradation in Cancer

Altered miRNA degradation represents yet another mechanism of miRNA dysregulation in cancer (Figure 5B). Although miRNAs are generically stable molecules, stability can be altered by several intracellular and extracellular cues. For rapid and drastic changes in miRNA expression, the molecules can be cleaved and degraded by endo- and exonucleases. Human polynucleotide phosphorylase (hPNPaseold-35) is a 3′, 5′ exoribonuclease that can be upregulated in human melanoma cells [174]. It is able to selectively degrade mature oncogenic miR-106, miR-221 and miR-222 in human melanoma cells [175]. In addition, high-complementarity of target mRNA is also able to induce miRNA degradation. This phenomenon is called TDMD and induces tailing and trimming of the miRNA, as explained previously [120]. One study tried to identify the role of TDMD in human cancer, using a computational predictive pipeline on a cancer dataset of thousands of samples [153]. They found well-known oncomiR and TS-miRs predicted to be targeted by TDMD. However, the consequences of this mechanism in cancer has not been elucidated yet.

### 7.12. Factors Affecting miRNA Editing in Cancer

miRNA dysregulation can also be a consequence of altered RNA editing. Since miRNA-induced gene expression regulation is based on complementarity between miRNA and target, post-transcriptional editing of the (im)mature miRNA sequence can change the targetome and alter the target protein expression in the cell (Figure 5C) [132]. When RNA is edited, this mainly involves the adaptation from adenosine to inosine by ADAR. Wang et al., performed a systematic analysis of miRNA editing patterns in 20 different cancer types [154]. They identified specific A to I editing hotspots, which are promising driver candidates in the development of cancer. One modification was found in miR-200b, where editing of its seed region resulted in a shift from a tumour suppressor to an oncogenic function. Higher miR-200b editing levels were associated with epithelial to mesenchymal transition (EMT) and with shorter OS in multiple cancer types. Moreover, Pinto et al. systematically characterized miRNA editing across 32 cancer types and normal controls [210]. The authors observed global hypo-editing of miRNAs in tumours, coupled with elevated mRNA editing. Editing of mRNA in the 3′ UTR also led to changes in the targetome of miRNAs. In many cancers, higher miRNA editing levels correlated with longer patient survival compared to those with low editing levels.

Another change that can be made in miRNA sequences in cancer is N6-methyladenosine (m6A) modification [211]. These modifications can be placed on pri-miRNA by MAC consists of methyl transferase 3 (METTL3), which promotes processing of the pri-miRNA through enhanced DGCR8 recognition. Expression of METTL3 is upregulated in bladder cancer, leading to increased maturation of miR-221/222 [176]. Upregulation of these miRNAs levels leads to downregulation of the target gene PTEN, which promotes proliferation in bladder cancer cells.

### 7.13. Dysregulation of miRNAs by Long Non-Coding RNA

As explained previously, lncRNAs can function as miRNA sponges. This mechanism of regulation can be altered in cancer. For example, expression of lncRNA HOX transcript antisense RNA (HOTAIR) can be upregulated in oesophageal cancer, which is associated with a poorer OS [177]. This lncRNA can act as a miR-148a sponge, to positively regulate Snail2 expression, promoting epithelial mesenchymal transition. Another example is the tumour suppressor lncRNA SLC25A5-AS1, which is significantly downregulated in gastric tumours [178]. The lncRNA is known to be associated with reduced proliferation, cell cycle progression and increased apoptosis in gastric cancer. SLC25A5-AS1 has distinct targets involved in cell cycle and growth regulation, among which miR-19a-3p was identified. This miRNA targets the PTEN mRNA, which negatively regulates the PI3K/AKT signalling pathway. As a result of reduced SLC25A5-AS1, miR-19a-3p expression is increased, as are its regulatory features, thereby promoting tumourigenesis in gastric [178] and breast cancer [179], among others.

### 7.14. Dysregulation of miRNAs by Circular RNA

Another sponge-like regulator of miRNA expression is the group of circRNAs. An example of a circRNA involved in cancer is ciRS-7, which contains more than 70 conserved miR-7 binding spots and thereby decreases its intracellular availability [181]. This impairs suppression of its targets, such as EGFR [182]. ciRS-7 was found to function as an oncogene and play a key role in various cancer types, including lung cancer, hepatocellular carcinoma, melanoma, colorectal cancer, breast cancer, and many others [183]. The link between cancer and circRNA dysregulation has been established in some instances, of which one is ciRS-7. However, other examples are sparse and not as obvious as ciRS-7 circRNA, which has an extraordinary amount of target binding sites (>70). Most circRNA transcripts contain less than ten miRNA binding sites, which does not make them very efficient sponges [212]. The lack of studies on this topic underlines that not much is known on how circRNA is dysregulated and how strongly this influences miRNA expression and cancer hallmarks. Additionally, more research needs to be invested into the positive effects of circRNA on miRNA expression, via increased stability for example [45].

### 7.15. Dysregulation of miRNAs by Transfer RNA Derived Fragments

When tRNA molecules are produced, they do not instantly fold into the right conformation. Instead, a chaperone protein called La is required to obtain the desired shape [213]. However, when La is absent, the pre-tRNA can misfold into several shapes, as a result of high complementarity between its 5′ and 3′ end. When this molecule is processed by RNaseZ, a tsRNA is formed that is known to compete with global miRNA-induced regulation by interacting with AGO proteins. This link between miRNA regulation and tRNAs could explain why tRNA levels are enriched in cancer [141]. On the other hand, when the molecule is processed by Dicer, normal and functional miRNAs can be produced [142]. For example, miR-1280 is obtained from its pre-miRNA, but also from the tRNA^Leu^ [184]. This molecule decreases Notch signalling, through directly targeting the Notch ligand JAG2. This leads to reduced tumour formation and metastasis in colorectal cancer.

### 7.16. Self-Regulation of miRNA Expression

The expression of a target precursor or mature miRNA can be regulated by other miRNAs in both direct and indirect ways. Direct interaction is for example seen between miR-424 or miR-503 and pri-mir-9 [185]. The two mature miRNAs promote differentiation, whereas miR-9 has a contradictory effect. This way, cell lineage commitment is stimulated. If this interaction is disrupted, miR-9 is upregulated, leading to an undifferentiated state typical of cancer cells. Since pri-miRNAs are produced and processed in the nucleus, regulation at this level requires the transport of mature miRNAs from the cytoplasm to the nucleus [214]. In addition to regulation by binding to pri-miRNA transcripts, mature miRNAs have also been reported to directly interact with other mature miRNAs. For example, miR-107 can bind to and suppress tumour suppressive miRNA let-7, which eventually leads to downstream target upregulation and oncogenesis [186].

Furthermore, miRNAs are also able to target the expression of other miRNAs indirectly, either by targeting transcription factors or the biogenesis machinery. The miR-20a targets the 3′ UTR of E2F transcription factors, resulting in downregulation of E2F expression [145]. One of the target genes of E2F, is miR-20a itself, creating an autoregulatory loop. Since E2F also has other target genes, this feedback mechanism can be seen as a way to prevent induction of apoptosis by E2F in for example prostate cancer cells. An example of a miRNA that can influence the biogenesis machinery is that of miR-98-5p upregulation in epithelial ovarian cancer [187]. This miRNA can target the mRNA transcript of Dicer, which induces global miRNA downregulation. Specifically, miR-152 levels were decreased, which induced chemotherapy resistance.

## 8. miRNAs as Cancer Biomarkers: Potential and Challenges

miRNAs have been found to be dysregulated in many cancer types and stages through various mechanisms. This includes TS-miRs and oncomiRs that can inhibit or promote important cancer hallmarks, respectively [8]. Since miRNAs play an important role in the development and progression of cancer, they have been proposed as early detection, diagnostic, monitoring, prognostic as well as predictive biomarkers in this context [10]. miRNAs can possibly serve as non-invasive biomarkers, since they are present in multiple biofluids at a relatively stable level [11]. miRNAs with diagnostic, predictive or prognostic value in clinical research have been extensively reviewed by Sempere et al. [215].

Currently, most biomarkers used in the clinic are proteins [216]. However, miRNAs are also suitable for multiple reasons. First, these molecules are relatively stable. When compared to mRNA, the half-life is on average almost 150 times higher [57,116]. This is mainly since they are resistant to degradation by ribonucleases, when present in exosomes, MVs, apoptotic bodies or bound by proteins [110,118,119]. Moreover, in contrast to mRNA, miRNAs are barely affected by repeated freeze and thaw cycles and are quite stable at room temperature [217]. This makes the handling of the molecules a lot less complex. Second, miRNAs are easy to extract and detection methods are sensitive [216]. Third, the expression of miRNAs is specific, indicating that specific miRNA levels can be linked to specific cancer types.

Although circulating miRNAs serve as a promising biomarker in cancer, there are multiple challenges that have to be overcome before being able to use them in the clinic [218]. One challenge is the fact that their origin remains unknown. miRNAs in blood can originate from all organs in the body. Moreover, if the expression profile in serum/plasma is measured, it is important to realise that miRNAs from red blood cells after haemolysis may obscure measurements [219]. Such a possible confounder should be taken into account when using circulating miRNAs as a biomarker. Furthermore, several systematic reviews have shown an abundance in contradictory results about miRNA expression levels in biofluids [220,221]. This might be the result of bias in sample processing, detection methodology, standardisation and normalization [218]. It is therefore difficult to compare studies and to determine what miRNA changes are related to cancer, and which are related to a differential experimental set-up. More standardized research has to be performed to accurately identify what sets of miRNAs are altered in specific cancer types and at what level these changes become significant.

Besides the technical challenges discussed above, there are also other variables that should be taken into account when interpreting circulating miRNA biomarker readouts. One important factor is the variability in miRNA levels between individuals [218]. These differences can be present based on age, gender or race, but can also be due to life-style differences like smoking, hormonal status, diet, alcohol and drug use [218,222]. Moreover, miRNA levels also change within the same individual over time, which makes it even more complex to determine specific levels for biomarker detection [218].

## 9. Conclusions and Future Directions

In this review, a comprehensive overview is provided on the regulatory mechanisms controlling miRNA biogenesis, availability and target interaction. It is important to understand the natural turnover of miRNAs to be able to use them as disease biomarkers in clinical practice.

A tumour is able to convert these miRNA regulatory mechanisms to its own benefit, decreasing the expression of TS-miRs and increasing the expression of oncomiR [8]. This can be obtained through many different regulatory mechanisms and feedback loops, making the dysregulation of miRNAs in cancer extremely complex [13]. miRNAs might have a potential future role in therapeutics by restoring TS-miR expression levels or sequestering oncomiR levels, although the regulatory complexity makes this difficult [223].

A few aspects of the miRNA lifecycle need to be further elucidated. For example, the effect of ceRNA on miRNA function in healthy cells is still poorly understood [224]. Fundamental studies conducted into the working mechanism of ceRNA-mediated miRNA regulation should be followed by the identification of ceRNAs that play significant roles in cancer. These ceRNAs could serve as reporters for dysregulated miRNA expression and might provide a model for the production of synthetic ceRNAs that exclusively target specific miRNAs [225]. Additionally, their role as miRNA sponges and competitors might prove valuable for a novel experimental approach, which uses ceRNAs to deplete miRNAs from the cells and study the effect of this dysregulation [224]. A similar approach might be taken to target oncomiR overexpression in a therapeutic setting, to treat tumours that depend strongly on their miRNA dysregulation.

Another gap in current understanding concerns the scale of miRNA dysregulation in cancer. It is known that altered miRNA expression supports tumour promoting behaviours on several occasions, but the extent of miRNA dysregulation in cancer is unclear [226]. Additionally, it is uncertain whether miRNA dysregulation is a facilitator, driver or supporter of tumourigenesis. Studies with large experimental groups should be conducted to determine specific miRNA level dysregulation in cancer, to reveal their targets and investigate their place within the dysregulated pathways of cancer. One final future direction involves the further elucidation of the miRNA-induced silencing and activation mechanisms. This review has covered both translational and transcriptional repression and activation, and whereas the repressive roles of miRNAs are extensively researched, miRNA-induced activation is a fairly recent discovery [65,227]. Fundamental studies should be conducted into the working mechanisms of this phenomenon, and subsequently, its role within cell homeostasis should be investigated. These findings could then be used to elucidate the role of this enhanced gene expression in cancer, which might provide new insights into therapeutical approaches.

In conclusion, miRNA function can be affected at three fundamental levels: miRNA production, miRNA availability and interaction of miRNAs with the target mRNA. All sub-processes influence the final production and function of miRNAs, and therefore influence miRNA target expression. In cancer, miRNAs can be dysregulated to alter tumour suppressor and oncogene regulation, promoting tumour development and progression. The exact contribution of miRNA dysregulation to tumourigenesis remains largely unknown, but current studies are revealing the roles of specific miRNAs in cancer and report the pathways on which they act. In combination with the elucidation of regulatory mechanisms controlling miRNA expression, these developments pave the way for a broader application of miRNAs, as biomarker or even therapeutic approach.

## Figures and Tables

**Figure 1 cancers-14-05748-f001:**
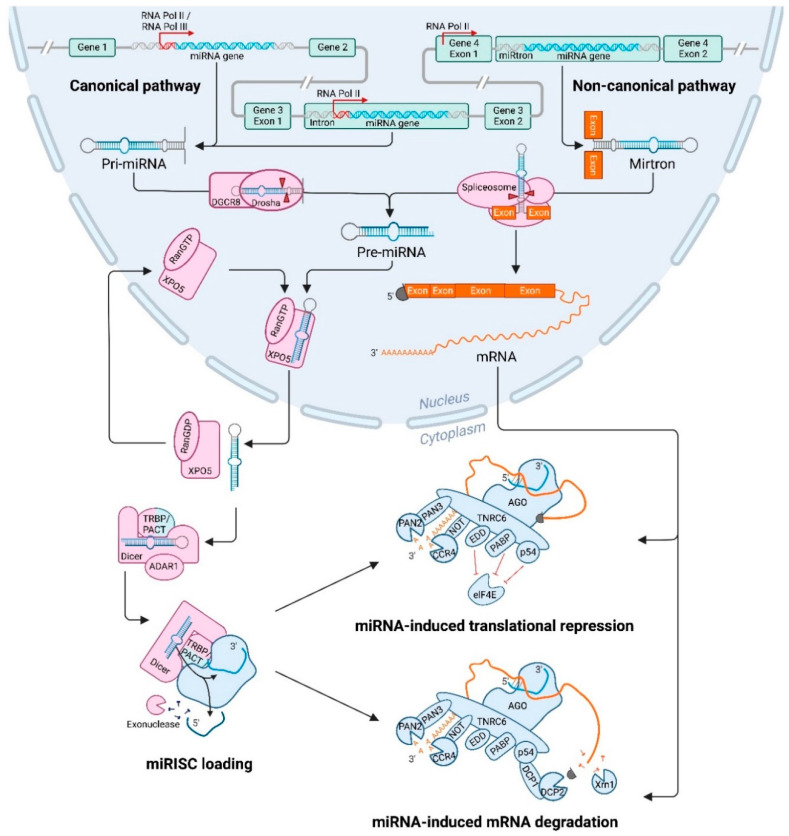
General overview of the canonical and non-canonical route from microRNA gene to regulation of target messenger RNA. Following the canonical pathway of miRNA biogenesis, an intergenic (left), or intragenic (middle) miRNA gene is transcribed by RNA polymerase (Pol) II or III, to a primary-miRNA (pri-miRNA) transcript. This stem-loop transcript is cleaved by ribonuclease III enzyme Drosha and its co-factor DiGeorge Syndrome Critical Region 8 (DGCR8) to form a precursor-miRNA (pre-miRNA), which is subsequently exported out of the nucleus by Exportin-5 (XPO5) and its co-factor Ran-GTP. The non-canonical pathway of miRNA biogenesis involves the splicing of a pre-miRNA molecule from an entire intron without the need for Drosha processing, after which it enters the canonical pathway. In the cytoplasm, the pre-miRNA is cleaved by ribonuclease III enzyme Dicer and its co-factors transactivation-response element RNA binding protein (TRBP), protein kinase RNA activator (PACT) and adenosine deaminases acting on RNA 1 (ADAR1) to form a mature miRNA duplex. After strand selection, a single-stranded guide miRNA is incorporated into Argonaute (AGO), which is bound by trinucleotide repeat containing 6 protein (TNRC6) and then the functional miRNA-induced silencing complex (miRISC) is formed. The passenger strand is released and degraded by an exonuclease. After association with multiple co-factors and miRISC effector proteins, the miRISC is able to silence its target. The miRNA is hereby used as a guide, and based on complementarity between miRNA and messenger RNA (mRNA), translation is repressed or the mRNA is degraded. Binding of the miRISC to the target mRNA can induces deadenylation by CCR4-NOT as main effector, which is in some cases followed by decapping by DCP1/DCP2 and degradation by exonuclease XRN1 (Created with BioRender.com).

**Figure 2 cancers-14-05748-f002:**
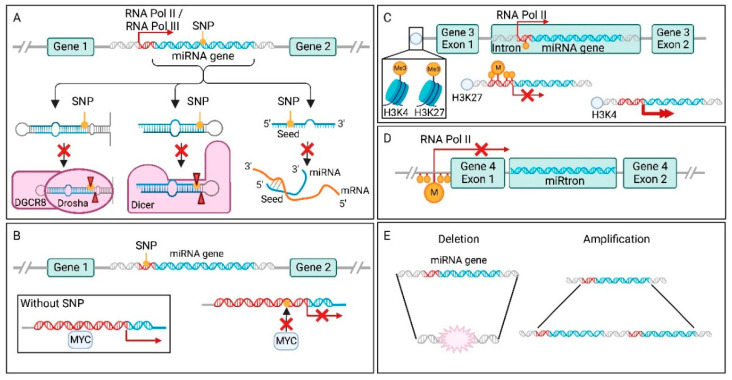
Genetic factors affecting microRNA expression and function in cancer. (**A**) Single nucleotide polymorphisms (SNPs) affecting the microRNA (miRNA) gene can alter miRNA production through impeding processing by ribonuclease III enzyme Drosha or Dicer, by alteration in binding and cleavage sites (red triangles). SNPs can also change the seed sequence, leading to a different targetome for the miRNA and thus altered post-translational regulation. (**B**) In addition to mutations in the miRNA gene itself, the promoter and transcription-enhancing region of miRNA genes can also acquire SNPs. This can impair binding of transcription factors, such as MYC, leading to decreased transcription of the miRNA gene. (**C**) Epigenetic modifications of the miRNA gene can also affect miRNA expression. Hypermethylation of the miRNA promoter, as well as histone methylation at H3K27, both repress transcription. On the other hand, histone methylation at H3K4 is associated with transcriptional activation of miRNA genes. (**D**) Some miRNA genes do not have a promoter of their own, which makes them dependent on their host gene. When the host gene promoter is hypermethylated, expression of the miRNA/mirtron is decreased as well. (**E**) Copy number variations are large chromosomal changes, which include deletions of miRNA genes, or translocations to other positions in the genome, creating gene amplification. When a part of the chromosome is amplified or deleted, the level of miRNA gene expression is decreased or increased, respectively (Created with BioRender.com).

**Figure 3 cancers-14-05748-f003:**
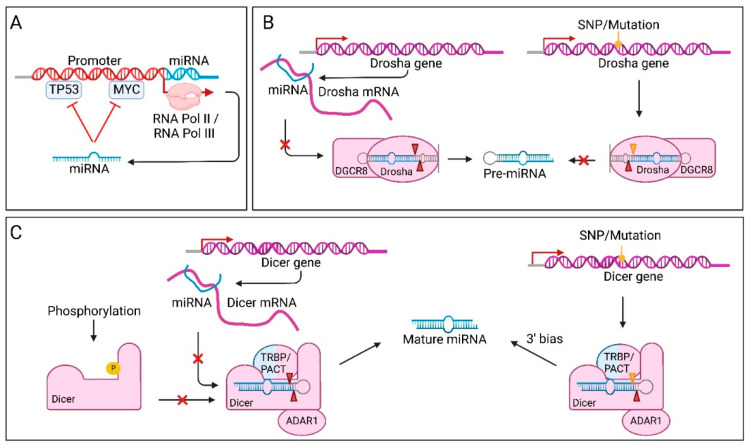
The effect of altered transcription and altered processing through ribonuclease III enzymes Drosha and Dicer malfunctioning. This figure shows how steps of the biogenesis affect production of microRNA (miRNA). (**A**) Often, miRNAs are transcribed by RNA Polymerase II, which requires transcription factors, such as TP53 and MYC. These transcription factors can be targets of the transcribed miRNA, resulting in a feedback loop. (**B**) The Drosha RNase III enzyme processes primary miRNAs to precursor miRNAs (pre-miRNA). The protein’s functionality can be altered by many factors, such as single nucleotide polymorphisms (SNPs) and mutations. Drosha messenger RNA can also be the target of specific miRNAs, leading to a negative feedback loop in miRNA maturation. (**C**) The Dicer RNase III enzyme is able to process pre-miRNA to form a mature miRNA duplex. SNPs and mutations in the Dicer gene can alter its function, leading to altered pre-miRNA processing. Mutations in the RIIID domain of Dicer lead to problems with 5′ strand processing, inducing a 3′ strand bias. Besides inhibition, some SNPs are also capable of increasing Dicer effectivity. Dicer can also be regulated by specific miRNAs, leading to lower miRNA maturation. Phosphorylation of Dicer can induce nuclear translocation, leading to altered miRNA processing in the cytosol (Created with BioRender.com).

**Figure 4 cancers-14-05748-f004:**
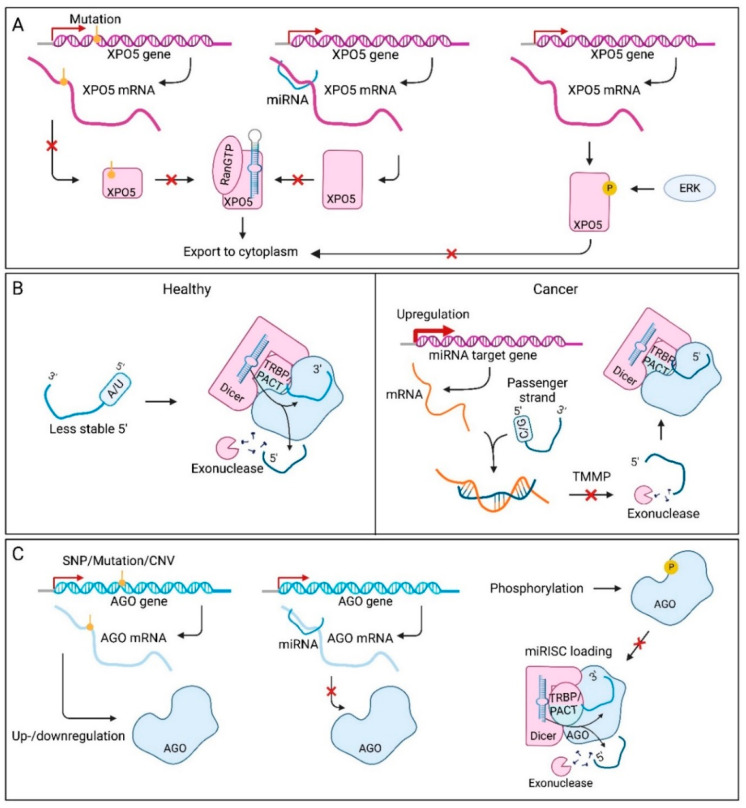
Defective microRNA (miRNA) export, miRNA-induced silencing complex (miRISC) loading and Argonaute (AGO) malfunction, leading to altered miRNA functionality in cancer. (**A**) Exportin-5 (XPO5) exports precursor miRNAs (pre-miRNAs) from the nucleus to the cytoplasm. When the XPO5 gene is mutated, pre-miRNA is retained in the nucleus, resulting in lowered mature miRNA levels. XPO5 functionality can also be altered through post-translational regulation by specific miRNAs. Moreover, phosphorylation by extracellular signal-regulated kinase (ERK) inhibits the shuttling of XPO5 to the cytosol, leading to decreased levels of pre-miRNA in the cytoplasm. (**B**) Loading of one of the miRNA strands into the miRNA-induced silencing complex (miRISC) determines targetome that will be regulated. Strand selection is determined by least 5′ end thermodynamics and preferably adenine or uracil nucleotides. The strand that meets these requirements the most, is implemented, whilst the other is degraded. However, when the interaction of the discarded strand with its own target mRNA is strong, this interaction prevents miRNA degradation, leading to arm switching. This mechanism is called target-mediated miRNA protection (TMMP). (**C**) The assembly of the miRISC can also be regulated through differences in AGO proteins. Transcription of the AGO genes can be altered through single nucleotide polymorphisms, mutations or copy number variations (CNVs) in cancer, leading to de- or increased miRNA functionality. AGO can also be targeted by specific miRNAs, creating a negative feedback loop. Moreover, the AGO proteins can be phosphorylated, leading to impaired miRISC loading (Created with BioRender.com).

**Figure 5 cancers-14-05748-f005:**
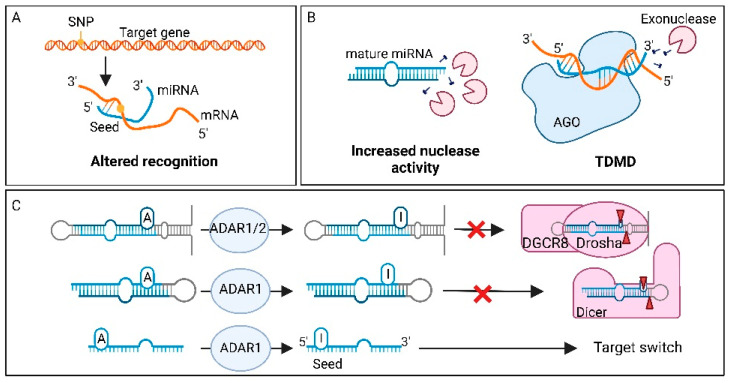
microRNA dysregulation in cancer through target gene modifications, and factors affecting microRNA degradation and microRNA editing. (**A**) A mutation in the microRNA (miRNA) recognition site of a target gene can result in multiple changes; the target can become undetectable for the miRNA, the mutation might change the degree of repression or make a target recognizable for miRNA. (**B**) miRNA stability in the cell is important for regulation of miRNA targets, and stability is lowered by increased nuclease activity or target-directed miRNA degradation (TDMD). The latter process, TDMD, revolves around complementarity between target and miRNA. When a miRNA binds to its target with high 3′ end complementarity, the conformation of Argonaute (AGO) may change and the miRNA may become susceptible to degradation by exonucleases. (**C**) RNA editing, such as adenosine (A) to inosine (I) editing by adenosine deaminases acting on RNA 1/2 (ADAR1/2), can change the sequence of the RNA, with diverse consequences. miRNA editing can alter primary-miRNA and pre-miRNA in such a way, that they become unavailable for processing by ribonuclease III enzymes Drosha and Dicer, respectively. Additionally, the seed can also be edited, which allows for a shift in targetome, or altered recognition of existing targets (Created with BioRender.com).

**Table 1 cancers-14-05748-t001:** Examples of dysregulated microRNAs in cancer. The first column indicates the step within the microRNA (miRNA) lifecycle that is altered, based on the chapters in this review. The second column includes the name of the miRNA(s) that have been altered. The third column describes the cause of the dysregulation, including genetic and post-translational changes. The fourth column explains the direct effect the miRNA alteration might have. The last column shows in what cancer type(s) this miRNA dysregulation has been identified, only including the types that have been described within this review. miRNA encoding genes are written in italics, while mature miRNAs are indicated with the capital letter R in miR.

Lifecycle Step	microRNA	Cause	Direct Effect	Cancer Type	References
Translational repression	*mir-499a-3p*	SNP	Altered targetome	Glioma	[147,159]
Processing of the pri-miRNA	*mir-26a*	SNP	Decreased pri-miRNA processing by Drosha	BLCA, OSCC, ESCC, LC, CC, BRCA, CRC	[160,161]
Processing of the pre-miRNA	*mir-196a-3p*	SNP	Increased pre-miRNA processing by Dicer	NSCLC	[162]
Transcription	*mir-378*	SNP	Decreased miRNA expression	BRCA	[148,162]
Transcription	*mir-130a*	Hypermethylation miRNA promoter	Decreased miRNA expression	PCa	[163]
Transcription	*mir-15/16*	CNV	Decreased miRNA expression	B-CLL, PCa	[164,165,166]
Transcription	*mir-29*	Upregulation transcription factor	Decreased miRNA expression	BLCA	[167]
Transcription	miR-4728	Host gene overexpression	Increased miRNA expression	BRCA, GC	[158]
Processing of the pri-miRNA	let-7	Drosha mutant	Decreased miRNA processing by Drosha	Wilms tumour	[155]
Processing of the pri-miRNA	miR-27b	Drosha SNP	Decreased post-translational repression of Drosha mRNA	BLCA	[146]
Processing of the pre-miRNA	miR-3622a-5p, miR-5582-5p	Dicer SNP	Increased post-translational repression of Dicer mRNA	TCL, HCC	[168,169,170]
Strand selection	miR-223	Altered NHEJ	Arm switching (3p to 5p)	BRCA	[171]
Strand selection	miR-193a	Increased expression target genes (TMMP)	Arm switching (5p to 3p)	BRCA	[172]
Translational repression	miR-187	SNP in target gene	Increased target mRNA repression	BRCA	[152,173]
Degradation	miR-106, miR-221/222	Increased expression exoribonuclease	Increased degradation	MEL	[174,175]
Modification	miR-200b	A to I editing	Altered targetome	HNSCC, KIRP, THCA, UCEC	[154]
Methylation	miR-221/222	m6A modification	Increased processing	BLCA	[176]
ceRNAs	miR-148a	Upregulation lncRNA	Decreased expression	ESCC	[177]
ceRNAs	miR-19a-3p	Downregulation lncRNA	Increased expression	GC, BRCA	[178,179,180]
ceRNAs	miR-7	Upregulation circRNA	Decreased expression	LC, HCC, MEL, CRC, BRCA	[181,182,183]
ceRNAs	miR-1280	Downregulation tRNA^Leu^ /miR-1280	Increased target mRNA repression	CRC	[141,184]
Self-regulation	miR-424, miR-503, mir-9	Disruption interaction between mature and pri-miRNA	Increased miR-9	AML	[185]
Self-regulation	miR-107 let-7	Upregulation miR-107	Degradation let-7	BRCA	[186]
Self-regulation	miR-20a	Upregulation miRNA	Negative autoregulatory loop through downregulation of transcription factor	PCa	[145]
Self-regulation	miR-98-5p	Upregulation miRNA	Global decrease in Dicer processing	OV	[187]

Abbreviations: bladder cancer = BLCA; oral squamous cell carcinoma = OSCC; oesophageal squamous cell carcinoma = ESCC; (non-small cell) lung cancer = (NSC)LC; cervical cancer = CC; breast cancer = BRCA; colorectal cancer = CRC; prostate cancer = PCa; B-cell chronic lymphocytic leukaemia = B-CLL; T-cell lymphoma = TCL; hepatocellular carcinoma = HCC; melanoma = MEL; head and neck squamous sarcoma = HNSCC; kidney renal papillary cell carcinoma = KIRP; thyroid cancer = THCA; endometrial cancer = UCEC; gastric cancer = GC; acute myeloid leukaemia = AML; ovarian cancer = OV; competing endogenous RNAs = ceRNAs; long non-coding RNA = lncRNA; circular RNA = circRNA; transfer RNA = tRNA; target mediated miRNA protection = TMMP; copy number variation = CNV; non-homologous end-joining = NHEJ.

## Data Availability

Not applicable.

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
