# Peer review of "The microRNA Lifecycle in Health and Cancer"

_cancers, 2022, doi:10.3390/cancers14235748_

Round 1

Reviewer 1 Report

The authors in the review “The microRNA lifecycle in health and cancer” have elaboratively described the miRNA life cycle. They have collected most of the information regarding miRNAs. This review is up to the mark for publication with some minor comments/additions.

1) There are other aspects of miRNAs (called small activating RNAs) which are very similar to miRNAs and have application in cancers too (28639200, 29625201) . I Suppose that is another important aspect of these small RNAs and should be included in this review.

2) Author should also add list/table of miRNAs that are used in clinical aspects or cilinical trials for the same topic especially cancer.

Thank you.

Best

Author Response

Thank you for reviewing our work. The following feedback has been changed in the revised manuscript: 

  • "There are other aspects of miRNAs (called small activating RNAs) which are very similar to miRNAs and have application in cancers too (28639200, 29625201) . I Suppose that is another important aspect of these small RNAs and should be included in this review."

We have now incorporated a sentence in chapter 3.3 (page 9), about the similarity between microRNAs and small activating RNAs within activation of transcription.

  • "Author should also add list/table of miRNAs that are used in clinical aspects or clinical trials for the same topic especially cancer."

Since we believe this is not within the primary scope of our review, we have now referred to another recent review that has extensively discussed this topic, in chapter 8 (page 26).

Reviewer 2 Report

This article describes the life cycle of microRNAs (miRNAs) in regulating gene expression at the post-transcriptional level in both health and cancer. In this review, the targets of miRNAs are described, indicating their essential function within physiological and pathological cellular processes. It describes miRNA production through canonical and non-canonical pathways involving a multitude of steps and proteins both in health and how they can be altered to promote tumor progression. The authors describe genetic causes, epigenetic changes, and differences in host gene expression or chromosomal remodeling, through an overview of current knowledge on the miRNA life cycle in health and cancer. The review is comprehensive and relates the various points of these complex pathways so that the topics covered so far on miRNAs can be found in all paragraphs. The manuscript is well-written, and the figures are well-explained. However, I would ask for tables to be introduced to bring the sections together to outline the chapters.

Author Response

Thank you for reviewing our work. The following feedback has been changed in the revised manuscript: 

  • "I would ask for tables to be introduced to bring the sections together to outline the chapters." A second reviewer also mentioned including a table, specifically for the information of chapter 7, which has now been added to the review, including references. Within this table, we have also added a first column that indicates which step of the microRNA lifecycle (which are chapter titles) is involved in the dysregulation. This will hopefully bring the sections together as was asked in this review. 

Reviewer 3 Report

This review aims to summarize the existing knowledge about the life cycle of microRNAs in health and cancer, covering  different aspects of microRNA biogenesis, function and regulation in normal, physiological conditions as well as cancer related alterations in these processes.

The review is focused on a relevant topic. The manuscript is written clear and presented in a well-structured manner.

Just minors:

1. The title of the Section 3 - "miRNA transcription" should be modified appropriately since this section describes miRNA functions. The same title already exists as a title of the subsection 2.1.

2.   The authors should consider summarizing some data in tables. For example, the mechanisms of miRNA dysregulation in cancer, with specific microRNAs known to be affected  in a particular way and cancer types where it was documented (with references).

Author Response

Thank you for reviewing our work. The following feedback has been changed in the revised manuscript: 

  • The title of the Section 3 - "miRNA transcription" should be modified appropriately since this section describes miRNA functions. The same title already exists as a title of the subsection 2.1.

This happened due to a copy/paste mistake, and has been changed to: miRNA target regulation. 

  • The authors should consider summarizing some data in tables. For example, the mechanisms of miRNA dysregulation in cancer, with specific microRNAs known to be affected  in a particular way and cancer types where it was documented (with references).

The information of chapter 7 has now been added to the review in a table, including references. Another reviewer also mentioned to outline the chapters within this table. This has been addressed in the first column of the new table, that indicates which step of the microRNA lifecycle is involved in the dysregulation.